# Polyphenols and Their Effects on Muscle Atrophy and Muscle Health

**DOI:** 10.3390/molecules26164887

**Published:** 2021-08-12

**Authors:** Takeshi Nikawa, Anayt Ulla, Iori Sakakibara

**Affiliations:** Department of Nutritional Physiology, Institute of Medical Nutrition, Tokushima University Graduate School, 3-18-15 Kuramoto-cho, Tokushima 770-8503, Japan; anayatullah1988@gmail.com (A.U.); sakakibara.iori@tokushima-u.ac.jp (I.S.)

**Keywords:** antioxidants, polyphenols, flavonoid, muscle atrophy, proteolysis, mitochondrial dysfunction, mitochondrial biogenesis, myogenesis, oxidative stress

## Abstract

Skeletal muscle atrophy is the decrease in muscle mass and strength caused by reduced protein synthesis/accelerated protein degradation. Various conditions, such as denervation, disuse, aging, chronic diseases, heart disease, obstructive lung disease, diabetes, renal failure, AIDS, sepsis, cancer, and steroidal medications, can cause muscle atrophy. Mechanistically, inflammation, oxidative stress, and mitochondrial dysfunction are among the major contributors to muscle atrophy, by modulating signaling pathways that regulate muscle homeostasis. To prevent muscle catabolism and enhance muscle anabolism, several natural and synthetic compounds have been investigated. Recently, polyphenols (i.e., natural phytochemicals) have received extensive attention regarding their effect on muscle atrophy because of their potent antioxidant and anti-inflammatory properties. Numerous in vitro and in vivo studies have reported polyphenols as strongly effective bioactive molecules that attenuate muscle atrophy and enhance muscle health. This review describes polyphenols as promising bioactive molecules that impede muscle atrophy induced by various proatrophic factors. The effects of each class/subclass of polyphenolic compounds regarding protection against the muscle disorders induced by various pathological/physiological factors are summarized in tabular form and discussed. Although considerable variations in antiatrophic potencies and mechanisms were observed among structurally diverse polyphenolic compounds, they are vital factors to be considered in muscle atrophy prevention strategies.

## 1. Introduction

The skeletal muscle is a plastic organ and the most abundant tissue in vertebrates. It plays a significant role in metabolism, movement, respiration, protection, daily physical activities, and the maintenance of posture and balance [1]. Generally, a healthy skeletal muscle always maintains a good equilibrium between protein synthesis and protein degradation. Any physiological (aging) or pathological conditions that interfere with the catabolic and anabolic symmetry of proteins will lead to a reduced cross-sectional area (CSA) of muscle fibers and decreased muscle strength and mass, resulting in muscle atrophy. The three main conditions that trigger muscle atrophy are (I) disuse, including immobilization, denervation, bed rest, space flight, and aging; (II) chronic disease, including chronic heart failure, obstructive lung disease, diabetes, renal failure, AIDS, sepsis, and cancer; and (III) medications, such as glucocorticoids [2,3]. Atrophy of the muscle reduces the quality of life and movement independence of the patients, thus imposing an additional financial burden on the health care system and causing increased morbidity and mortality. Hence, maintaining healthy muscles is necessary for preventing metabolic diseases and achieve healthy aging.

Muscle mass is maintained via the regulation of various anabolic pathways. Among them, the PI3K/Akt/mTOR pathway is the most important, in which insulin/IGF-1 acts as the upstream molecule to promote protein synthesis and block protein degradation [4]. The activation of Akt, a serine/threonine kinase, through phosphatidylinositol 3-kinase (PI3K) activates the mammalian target of rapamycin (mTOR) to induce protein synthesis by activating its downstream effectors, the ribosomal protein S6 kinase beta-1 (S6K1), and the eukaryotic translation initiation factor 4E (eIF4E)-binding protein 1 (4E-BP1) [4]. Activated Akt also phosphorylates the Forkhead box O (FoxO) transcription factors, leading to their expulsion from the nucleus, thereby preventing the transcription of genes encoding vital E3 ubiquitin ligases, such as muscle atrophy F-Box (MAFbx/atrogin-1) and muscle RING finger-1 (MuRF-1), or autophagy-related genes, such as those encoding the microtubule-associated protein 1A/1B-light chain 3 (LC3) and Bcl2/adenovirus E1B 19-kDa-interacting protein 3 (Bnip3), which are responsible for protein breakdown mediated by the ubiquitin–proteasome system (UPS) and autophagic–lysosomal pathway, respectively [4,5,6]. Various proatrophic factors, such as inflammatory cytokines (tumor necrosis factor-α (TNF-α), interleukin-6 (IL-6), and IL-1β), glucocorticoids, oxidative stress, and mitochondrial damage, were found to reduce Akt activation and induce muscle atrophy [7,8,9]. Among the protein catabolic pathways, the UPS is the chief proteolytic system, which degrades muscle proteins by upregulating the ubiquitin ligases atrogin-1, MuRF-1, and casitase-B-lineage lymphoma (Cbl-b). Generally, the UPS is activated by impaired PI3K/Akt signaling, inflammatory cytokines, oxidative stress, and mitochondrial dysfunction [9,10,11]. Furthermore, the nuclear factor-κB (NF-κB) transcription factor, which is an inducer of atrogin-1 and MuRF-1, is activated by inflammation and oxidative stress and initiates protein degradation via direct binding to the *MuRF-1* promoter [12]. Additionally, myostatin, a potent negative regulator of muscle growth and differentiation, induces muscle atrophy by inducing the expression of atrogin-1 and MuRF-1 [13].

Oxidative stress and inflammation are important factors strongly associated with muscle atrophy [14]. Oxidative stress can be defined as an imbalance between oxidants (ROS) and antioxidants in the body. Several conditions such as chronic diseases, cachexia, disuse, denervation, aging, etc. are associated with increased oxidative stress, that causes activation of proteolytic pathways as well as mitochondrial dysfunction [15]. The catabolic regulatory elements such as NF-kB, FoxOs, AMPK, etc. are activated by oxidative stress that induce muscle protein catabolism by regulating UPS. It also induces the ubiquitin ligase Cbl-b, which disturbs IGF-1 signaling by ubiquitinating the insulin receptor substrate-1 (IRS-1), that leads to the activation of FoxOs and FoxOs-mediated ubiquitin ligases [16,17]. Inflammation triggers muscle atrophy during cachexia, chronic disease, aging, denervation, and obesity. Inflammatory cytokines (TNF-a, IL-6, IL-1b) are released during the above conditions, that interfere with the pathways associated with protein synthesis and proteolysis [18]. They activate various pathways including NF-kB and p38-MAPK as well as myostatin to induce proteolysis via UPS [19]. Polyphenols show strong antioxidant and anti-inflammatory effects; therefore, they could play a significant role in counteracting muscle atrophy induced by inflammation and oxidative stress (Figure 1).

Muscle atrophy is closely associated with mitochondrial quality. Mitochondrial quality and biogenesis are indispensable factors for proper muscle function, as impaired mitochondrial activities lead to muscle atrophy via the activation of various catabolic pathways [9,20]. Healthy mitochondria always maintain muscle homeostasis by producing adequate adenosine triphosphate (ATP) via the tricarboxylic acid cycle and oxidative phosphorylation (OXPHOS) [21]. They also regulate the antioxidant defense system and apoptosis (a type of programmed cell death). Moreover, the biogenesis of mitochondria is regulated by the peroxisome proliferator-activated receptor-gamma coactivator 1-alpha (PGC-1α), which activates nuclear respiration factors (Nrf1 and Nrf2) and the estrogen-related receptor-α, which further activates the mitochondrial transcription factor A (TFAM) to promote mitochondrial DNA (mtDNA) replication [21].

Furthermore, myogenesis plays a vital role in the preservation of muscle health and the attenuation of atrophy. Myogenesis is a process via which satellite cells are differentiated to myofibers to maintain muscle tissue regeneration. This process is regulated by various myogenic regulatory factors, namely, the myoblast determination protein (MyoD), myogenic factor 5 (Myf5), myogenin, and myogenic regulatory factor4, which lead to the determination and differentiation of skeletal muscle cells [22]. Hence, both healthy mitochondria and myogenesis are required for healthy muscle activities.

Polyphenols (PPs) are naturally occurring organic compounds that are abundantly found in different plants, fruits, vegetables, nuts, seeds, flowers, tea, and beverages [23]. PPs are worthier because of their diversity, bioactivity, easy accessibility, and the specificity of the response, with lower toxicity effects. However, rapid metabolism and low bioavailability are their major drawbacks [24]. PPs are classified mainly into four groups, i.e., (a) phenolic acids, (b) flavonoids, (c) stilbenes, and (d) lignans, depending on the number of phenol rings included in their structure and the structural components that bind these rings together [25]. Researchers have reported the wide-ranging health benefits of PPs, including the neuroprotective, cardioprotective, renoprotective, hepatoprotective, antidiabetic, anticancer, antioxidant, anti-inflammatory, and antimicrobial activities of PPs, which can be attributed to their diverse biological properties [24]. Thus, recently, PPs have drawn extensive attention among nutrition scientists regarding the exploration of their improved consumption as functional foods that counteract different diseases. Many of them have been trialed for their possible use as clinical treatments [26]. PPs have also been reported to exert strong antiatrophic effects by modulating the proatrophic factors or signaling pathways that contribute to muscle atrophy [27]. Because of their lower toxicity and higher target-specific response, PPs are being vigorously investigated as muscle atrophy countermeasures.

This review primarily appraises the studies of the benefits of PPs as potent bioactive molecules regarding muscle health, which include their roles in activating signaling molecules that are responsible for protein synthesis and the prevention of protein degradation via different pathways. It also sheds light on the role of PPs in enhancing mitochondrial function and myogenesis by improving mitochondrial quality and biogenesis and regulating myogenic regulatory factors. The induvial subclass of polyphenolic compounds is also discussed to elaborate on their role.

## 2. Methodology

To review the current information on the effect of PPs on muscle atrophy and muscle health, an in-depth search was conducted on PubMed and Google Scholar using the keywords or phrases “polyphenol in muscle atrophy”, “polyphenol in muscle dystrophy”, and “polyphenol in muscle damage or disorder”. Moreover, to include the effect of each subclass of polyphenol, we performed a further search using the keywords “phenolic acid in muscle atrophy”, “flavonoids in muscle atrophy”, “stilbenes in muscle atrophy”, and “lignan in muscle atrophy”. The closest synonymous words for atrophy, such as dystrophy, damage, disorder, and dysfunction, were also used in searches, in replacement of atrophy. Furthermore, references to specific items in the literature were carefully evaluated to obtain clear information. The information obtained for each subclass of polyphenol was summarized in tabular form and then discussed to explain their effects on muscle health and the underlying mechanisms.

## 3. Polyphenols in Managing Muscle Atrophy and Muscle Health

### 3.1. Phenolic Acids

Phenolic acids are important bioactive compounds of the polyphenolic group. They consist of an aromatic ring with several hydroxyl groups attached to it. They are primarily subclassified into two classes: hydroxybenzoic acids and hydroxycinnamic acids. Generally, hydroxybenzoic acid and hydroxycinnamic acid contain seven (C6-C1) and nine (C6-C3) carbon atoms, respectively, in their structure, with few exceptions. For example, compounds such as Ellagic acid of the hydroxybenzoic acid group contain 14 carbon atoms in their structure. Similarly, Chlorogenic acid of hydroxybenzoic acid group has 16 carbon atoms in its structure. Most commonly they are distributed in cereals: fruits, such as pomegranates, apples, grapes, raspberries, strawberries, cranberries, blackcurrants, and walnuts, and vegetables [28]. Pharmacologically, they exhibit antioxidant, anti-inflammatory, neuroprotective, cardioprotective, hepatoprotective, antidiabetic, anticancer, and antimicrobial effects [28]. Some representative phenolic acids have been reported to exert beneficial effects on muscles by promoting their growth and/or reducing their wasting while improving mitochondrial quality and preventing inflammation and oxidative stress, as summarized in Table 1.

Hydroxybenzoic acids include gallic acid, vanillic acid, ellagic acid, salicylic acid, and protocatechuic acid. Few studies have reported the effects of these compounds in skeletal muscle. Recently, Chang et al. (2021) reported that gallic acid improved mitochondrial functions in C2C12 myotubes by activating SIRT-1, which in turn activates the transcription factors PGC-1α, Nrf1, and TFAM, to promote mitochondrial biogenesis. It also upregulated mitochondrial biogenesis, oxidative phosphorylation, myosin heavy chain (MyHC) content, autophagy/mitophagy, and the fusion/fission index of mitochondria in C2C12 myotubes [29]. Gallic acid (GA) in combination with epicatechin (EC) and epigallocatechin (EGC) increased muscle differentiation by inducing the myogenic regulatory factors myogenin, Myf5, and MyoD in C2C12 myotubes [30]. Another compound of this class, ellagic acid, exerts antioxidant, anti-inflammatory, anticancer, hypolipidemic, and neuroprotective effects [31]. Ellagic acid treatment protected muscle tissue in rats against CCL4-induced muscle damage by reducing oxidative stress and inflammation. Moreover, it downregulated malondialdehyde (MDA), NF-κB, TNF-α, and cyclooxygenase-2 (COX-2) and upregulated antioxidant enzymes (catalase and glutathione [GSH]), together with its transcriptional factor nuclear factor erythroid 2-related factor 2 (Nrf2). Furthermore, suppressing B-cell lymphoma 2 (bcl-2) and inducing caspase-3 led the damaged tissues to apoptosis [32]. It has also been shown to improve muscle dysfunction in cuprizone-induced demyelinated mice by imparting mitochondrial protection, oxidative stress prevention, and Sirt3 overexpression [33].

Urolithins are metabolites obtained from the transformation of ellagic acid. Urolithin A and urolithin B have positive effects on muscle cells. Urolithin A, a mitophagy activator, stimulated muscle function and exercise capacity by inducing autophagy and mitophagy in rodents [34]. In a mouse model of Duchenne muscular dystrophy (DMD), it enhanced muscle function and survival rate by preserving mitophagy, respiratory capacity, and muscle stem cell regeneration ability. Mitochondrial dysfunction, which is the causative agent of DMD, was prevented via urolithin A treatment [35]. Moreover, skeletal muscle cell angiogenesis was increased by urolithin A through an increase in ATP and NAD+ levels. Angiogenesis was upregulated by urolithin A via the Sirt1–PGC-1α pathway [36]. Urolithin B induced the growth and differentiation of C2C12 cells, as well as hypertrophy, in denervation-induced muscle in mice. It decreased muscle atrophy by activating protein synthesis and inhibiting protein degradation. The anabolic activity of urolithin B was attributed to the activation of the androgen receptor, which activates the mTOR pathway. Additionally, it suppressed the upregulation of ubiquitin ligases, MAFbx/atrogin-1, MuRF-1, and myostatin [37]. Pomegranate, which is a rich source of ellagic acid, protected tibialis anterior muscle loss in TNF-α induced muscle atrophy. This protective effect was mediated by the suppression of the inflammatory cytokines, TNF-α, IL-1β, monocyte chemoattractant protein 1 (MCP-1), and NF-κB signaling-mediated ubiquitin ligases; however, it preserved the Akt/mTOR protein synthesis pathway [38].

Among the various hydroxycinnamic acids, ferulic acid (FA), chlorogenic acid, and caffeic acid have been reported to have positive effects on muscle. In mouse C2C12 myotubes, ferulic acid increased the MyHC protein (MyHC-I and MyHC-IIa) and decreased fast-type MyHC-IIb expression. The mRNA expression of sirtuin1 (Sirt1), PGC-1α, and myocyte enhancer factor 2C (MEF2C) was induced by FA. These effects were mediated by the phosphorylation of AMPK, followed by Sirt1 activation, as inhibition of AMPK or Sirt-1 abolished the effects of FA [39]. In another study, the administration of FA promoted the growth of both fast glycolytic and slow oxidative muscle in dexamethasone (Dex)-induced myopathic rats by downregulating myostatin and oxidative stress. FA induced the expression of mechano growth factor and enhanced the antioxidant enzymes superoxide dismutase (SOD), GSH, and catalase in the tibialis anterior (TA) and soleus muscles [40]. In a zebrafish model, FA administration for 30 days caused the hypertrophic growth of fast-type skeletal muscle via the phosphorylation of zebrafish target of rapamycin, p70S6K, and 4E-BP1. Moreover, the myogenic transcription factors myogenin, MyoD, and serum response factor were increased via FA treatment [41]. Chlorogenic acid has been reported to increase muscle strength by enhancing mitochondrial function and cellular energy metabolism in resistance training-induced rats [42]. Caffeic acid, which is found in fruits, wine, and coffee, exhibits some interesting effects against muscle injury. The administration of caffeic acid phenethyl ester (CAPE) protected muscles against eccentric exercise-induced injury in rats. This effect was mediated by the blockage of NF-κB activation and NF-κB-mediated prooxidant and proinflammatory responses. Inflammation-related genes, i.e., *IL-β*, *MCP-1*, *iNOS*, and *COX-2*, and the oxidative stress marker MDA were significantly suppressed via CAPE treatment [43]. Caffeic acid in combination with curcumin improved autosomal recessive spinal muscular atrophy (SMA) by promoting *SMN2* gene expression, which controls the severity of SMA. Etiologically, SMA is caused by a loss of α-motor neurons in the anterior horn of the spinal cord, leading to neurogenic muscle atrophy [44]. Moreover, the consumption of coffee, a good source of caffeic acid, has been reported to increase skeletal muscle function and hypertrophy by increasing MyHC, MyHC-IIa, and MyHC-IIb in the quadriceps muscle. It suppressed transforming growth factor-beta (TGF-β) and myostatin expression and upregulated the levels of the insulin-like growth factor-1 (IGF-1), and subsequently phosphorylated Akt and mTOR. It also enhanced PGC-1α expression, which is vital for myogenic differentiation. Hence, coffee exerted a protective effect by regulating the TGF-β/myostatin–Akt–mTORC1 pathway [45]. However, P-coumaric acid, of the phenolic acid group, reduced the differentiation of skeletal muscle cells by downregulating myogenin and MyoD [46].

### 3.2. Flavonoids

Flavonoids are bioactive polyphenolic compounds that are found in almost all fruits and vegetables. They are the most studied natural compounds for their valuable pharmacological effects in different diseases. Approximately 6000 flavonoid compounds are estimated to be distributed in different fruits, vegetables, herbs, and plants [47]. Flavonoids offer diverse health benefits against chronic diseases, including cardiovascular disease, kidney disease, cancer, diabetes, obesity, and hepatic disease, which lead to increased mortality and morbidity [48,49,50]. Because of their potent antioxidant and anti-inflammatory activities, they can modify various enzymes and signaling molecules that are responsible for disease prognosis [51]. Numerous in vitro and in vivo studies have reported flavonoids as potential therapeutic countermeasures for treating muscle atrophy or muscle injury via different mechanisms. The effects of each subclass of flavonoid, i.e., flavanols, flavanones, flavones, isoflavones, flavonols, and anthocyanin, have been summarized in Table 2 and are discussed below.

### 3.3. Flavanols

Among the three main compounds of the flavanol group, EC and epigallocatechin gallate (EGCG) have been extensively studied regarding their effects in muscle atrophy. In aged mice and human muscle samples, EC administration reversed the aging-induced expression of myostatin and β-galactosidase while increasing the level of follistatin, MyoD, and myogenin, which are the growth and differentiation factors of muscles [52]. Moreover, supplementation with EC for 37 weeks exerted an antiaging effect and increased the survival of mice from 39% to 69%, while attenuating aging-induced muscle loss. Furthermore, it prevented the aging-induced decline in the metabolism of nicotinate and nicotinamide, which play a vital role in mitochondrial respiration and oxidative phosphorylation [53]. EC counteracted the wasting of oxidative muscle (slow-type MyHC) in disuse-induced mice by increasing angiogenesis in muscle and improving mitochondrial function. It also decreased Foxo1 and angiogenic inhibitor thrombospondin-1 (TSP-1), whereas it increased the PGC-1β, mTOR, and AKT proteins [54]. Similarly, in denervated rats, treatment with EC for 30 days prevented muscle wasting via the downregulation of the UPS. This effect was achieved by downregulating Foxo1, MAFbx/atrogin-1, MuRF-1, and protein ubiquitination [55]. In the Becker muscular dystrophic muscle, EC increased muscle regeneration by upregulating Myf5, MyoD, myogenin, MEF2a, and follistatin and downregulating myostatin. It acted as mitochondrial bioenergetics by promoting mitochondrial biogenesis through PGC1-α [56]. In C2C12 cells, EC treatment enhanced myotube growth and mitochondrial biogenesis by activating the transcription factors Nrf2, TFAM, and citrate synthase through G-protein-coupled estrogen receptor (GPER) activation [57]. EC also increased myogenic differentiation through the stimulation of the promyogenic signaling pathways p38MAPK and Akt. Moreover, it elevated MyoD activity as well as the myogenic conversion and differentiation of fibroblasts [58]. In high-fat (HF) diet-induced aged mice, EC attenuated muscle damage by reducing the expression of Foxo1 and MuRF-1. Moreover, it boosted physical performance and prompted the growth and differentiation factor MEF2A [59]. We reported previously that EC treatment suppressed the expression of the ubiquitin ligases atrogin-1 and MuRF-1 induced by 3D clinorotation in C2C12 myoblasts and myotubes via the dephosphorylation of extracellular regulated kinase (ERK) signaling [60].

EGCG is another component of the flavanol group that is mainly found in green tea. In a sarcopenic rat model, aged rats exhibited muscle atrophy associated with an increase in protein degradation and a decrease in anabolic factors compared with their younger counterparts. EGCG treatment suppressed muscle atrophy and preserved gastrocnemius muscle (GM) mass. Treatment with EGCG significantly reduced the ubiquitin–proteasome complex (19S and 20S), myostatin, and the E3 ubiquitin ligases atrogin-1 and MuRF-1 in aged rats, whereas it upregulated the anabolic factors IGF-1 and IL-15 [61]. Similarly, EGCG treatment attenuated age-related GM loss via Akt phosphorylation and subsequent inhibition of FoxO1a-mediated MuRF1 and atrogin-1 expression in aged senescence-accelerated mouse-prone 8 (SAMP8) mice and in late-passage C2C12 cells. These effects were mediated by the upregulation of miRNA-486-5p via EGCG. Thus, it inhibited myostatin/miRNA/ubiquitin–proteasome signaling [62]. In starvation- and TNF-α-induced C2C12 cells, EGCG treatment significantly suppressed protein degradation and activated protein synthesis. Additionally, it increased the phosphorylation of Akt and Foxo3a and attenuated the 20S proteasome, atrogin-1, and MuRF-1 [63]. These results are consistent with those of our previous study, in which EGCG decreased 3D-clinorotation-induced atrogin-1 and MuRF-1 expression by dephosphorylating ERK [60]. In Lewis lung carcinoma (LLC) tumor-bearing mice, EGCG attenuated muscle atrophy by suppressing NF-κB and its downstream mediators MuRF-1 and MAFbx [64]. Similarly, in the dystrophic mdx5Cv mouse model, EGCG protected the fast-twitch extensor digitorum longus (EDL) muscle from necrosis and stimulated muscle function, thus showing a positive effect on dystrophic abnormalities [65]. In disuse-induced aged rats, EGCG improved plantaris muscle weight and size during reloading followed by unloading by suppressing proapoptotic signaling in reloaded muscle [66]. In a similar model, EGCG induced autophagy during unloading, whereas it suppressed autophagy in the reloaded condition, to improve muscle health [67]. Moreover, EGCG administration suppressed MuRF1 expression in Dex-induced C2C12 cells through the activation of the 67-kDa laminin receptor (67LR) [68]. The same study reported the protection afforded by EGCG against GM loss in disused mice, which was further potentiated by the use of eriodyctiol [68]. EGCG prevented HF-induced muscle atrophy in SAMP8 mice by improving insulin resistance, together with a change in serum leukocyte cell-derived chemotaxin 2 (LECT2) [69]. Interestingly, in human skeletal muscle fibers, ECGC treatment had a similar effect, as did the insulin growth factor 1 (IGF-1) and insulin, thus suppressing the action of the atrophy-promoting transcription factor Foxo1 via the net translocation of Foxo1 out of the nucleus [70].

Catechin flavanol-activated satellite cells, which induce the Myf5 transcription factor via Akt phosphorylation. It also promoted myogenic differentiation by inducing the myogenic markers myogenin and muscle creatine kinase for muscle regeneration in C2C12 cells. The same study, which was performed using mice, reported increased fiber size and muscle regeneration after EGCG treatment [71]. Furthermore, it attenuated downhill running-induced muscle damage by suppressing muscle oxidative stress and inflammation and hastened the recovery of physical performance in mice [72]. Catechin also diminished unloading-induced contractile dysfunction and muscle atrophy in unloaded mice by suppressing oxidative stress [73].

### 3.4. Flavanones

Flavanones, also called dihydroflavones, are another class of flavonoid compounds that are most commonly found in citrus fruits, such as oranges, lemons, and grapes. Hesperidin, naringenin, and eriodyctiol are the main compounds in this group. They exhibit antioxidant, anti-inflammatory, antitumor, and cholesterol-lowering effects [111]. In aged mice, hesperetin reverted the aging-induced decrease in muscle fiber size and improved running performance. The same in vitro study using human skeletal muscle cells showed that hesperidin increased mitochondrial function by promoting ATP production and mitochondrial spare capacity. Moreover, it reduced oxidative stress in vitro and in vivo [74], proving that this compound is a potential agent for the retardation of sarcopenia and mitochondrial dysfunctionalities. Conversely, naringenin delayed the differentiation of skeletal muscle cells by hampering ERα-mediated Akt phosphorylation [75]. Intriguingly, we found that the chemical modification of naringenin to 8-prenylnaringenin (8-PN) by substituting the 8-prenyl group in the hydrogen atom of the eighth position of the molecule significantly alleviated denervation-induced loss of GM, whereas intact naringenin was found to be ineffective. Moreover, 8-PN, which acts as a powerful phytoestrogen, attenuated the induction of atrogin-1 by denervation and accelerated Akt phosphorylation. The accumulation of 8-PN in the GM was 10-fold higher than that observed for naringenin, which suggested that the prenylation of naringenin is required for its accumulation in the GM and for improving muscle atrophy [76]. Similarly, we reported that 8-PN treatment in mouse C2C12 skeletal myotubes increased the phosphorylation of PI3K, Akt, and p70S6K1, which was associated with its estrogenic activity. It also enhanced the weight of the tibialis anterior muscle in casting-induced muscle-atrophied mice [77].

### 3.5. Flavones

Flavones represent one of the important groups of flavonoids and are commonly found in fruits, vegetables, and plant-derived beverages, such as those from grapefruits, oranges, parsley, onions, tea, chamomile, and wheat sprouts [112]. Apigenin (AP), luteolin, tangeritin, and chrysin are the main compounds of this group. Among them, AP and luteolin have been highlighted for their antiatrophic effect in vitro and in vivo. AP administration prevented aging-induced loss of TA, EDL, and soleus muscle mass, and increased the muscle fiber size in aged mice. The attenuation of muscle atrophy was associated with a reduction of oxidative stress and the inhibition of hyperactive mitophagy/autophagy and apoptosis. AP increased SOD and GSH expression and enhanced mitochondrial function by increasing ATP levels and mitochondrial biogenesis by inducing the expression of PGC-1α, TFAM, Nrf-1, and ATP5B [78]. Furthermore, AP treatment ameliorated obesity-induced skeletal muscle atrophy and increased muscle mass, the CSA of muscle fibers, and exercise capacity. Moreover, it suppressed the expression of the atrophic genes *MuRF1* and *atrogin*-*1*. The expression of the inflammatory cytokines TNF-α, IL-6, and IL-1β was attenuated by AP in the serum and GM muscle tissue. The mitochondrial function was enhanced by AP through the upregulation of citrate synthase, complex-I, and complex-II activities, leading to increased OXPHOS. AP also caused increased mitochondrial biogenesis by upregulating PGC-1α, TFAM, cytochrome c, somatic (CyCs), and mtDNA content. The same study found that AP suppressed palmitic acid-induced muscle atrophy and mitochondrial dysfunction in C2C12 myotubes, and further reported that the beneficial effect of AP was regulated by AMPK activation [79]. In denervation-induced mice, AP attenuated the loss of GM and soleus muscle weight and increased muscle fiber CSA. Additionally, it upregulated the total MyHC protein and the mRNA expression of MyHC-IIb in the GM, as well as MyHC-IIa in the soleus muscle, with suppression of MuRF1 expression. Furthermore, TNF-α was downregulated both in the GM and soleus muscles, whereas IL-6 was decreased only in the soleus muscle [80]. Similarly, AP increased the mRNA expression of MyHC-I, MyHC-IIa, and MyHC-IIb in the quadriceps muscles of mice. The G-protein-coupled receptor 56 (GPR56) and its ligand, collagen III, were upregulated, as were PGC-1α, PGC-1α1, PGC-1α4, IGF1, and IGF2. The protein arginine methyltransferase 7 (Prmt7) was also enhanced by AP. The same in vitro study found that AP stimulated C2C12 myogenic differentiation by regulating MyoD protein expression. Thus, the beneficial effect of AP was regulated via the Prmt7–PGC-1α–GPR56 pathway, as well as by the Prmt7–p38–myoD pathway [81].

Previously, we reported that treatment with AP and luteolin prevents the lipopolysaccharides (LPS)-mediated reduction of myotube diameter by suppressing atrogin-1/MAFbx expression through the inhibition of the JNK signaling pathway in C2C12 myotubes [82]. Moreover, in cachexia-induced muscle atrophy, luteolin treatment protected against the loss of GM, which was associated with the attenuation of the expression of inflammatory cytokines (TNF-α and IL-6) and the ubiquitin ligases MuRF-1 and Atrogin-1. Here MuRF-1 expression was reduced via the deactivation of NF-κB both at the transcription and translation level, whereas atrogin-1 was suppressed by the downregulation of p-38 [83]. Luteolin also improved GM mass and strength in Dex-induced myopathy by exerting its antioxidant and antiapoptotic activities. Moreover, it reduced oxidative stress with attenuation of MDA and enhancement of the GSH antioxidant. Finally, it suppressed apoptosis by regulating caspase-3 expression [84].

### 3.6. Isoflavones

Isoflavones are flavonoid compounds that are most commonly present in soybeans, soy foods, and legumes. They act as phytoestrogen and exhibit pseudohormonal activity in conjugation with the ER. They possess antioxidant, anticancer, antimicrobial, and anti-inflammatory activities [113]. Genistein and daidzein are the representative elements of this group. Isoflavones have been reported to exert favorable effects on inflammation and glucocorticoid-induced muscle myopathies. We reported the effects of isoflavone treatment (daidzein and genistein) in muscles in vitro and in vivo. Isoflavone suppressed TNF-α-induced C2C12 myotube atrophy by attenuating MuRF-1 activity. This effect was regulated by AMPK-mediated SIRT-1 activation, in which SIRT-1 caused deacetylation of p65, which is involved in the activation of MuRF-1 [85]. Similar effects were noted in an in vivo model, where isoflavone supplementation in tumor-bearing mice attenuated the decrease in GM mass and myofiber size. Isoflavone decreased the expression of the ubiquitin ligases atrogin-1 and MuRF-1 by reducing the phosphorylation of ERK [86]. In denervation-induced muscle atrophy in mice, supplementation with the isoflavone aglycone (AglyMax) at a 0.6% dose significantly attenuated fiber atrophy by suppressing apoptosis-dependent signaling, as manifested by decreased apoptotic nuclei, but not caspase-3 [87]. In the same model of atrophy, genistein protected against soleus muscle atrophy by suppressing the transcription of FOXO1 via the recruitment of FOXO1 to ER-α. Sequentially, the activation of ER-α, attenuates FOXO1 and subsequently the expression of atrogin-1 and MuRF-1 [88]. In C2C12 myoblasts, genistein treatment at a lower concentration prompted myoblast proliferation and differentiation; however, at higher concentrations, it inhibited both of these processes. It also downregulated miR-222, therefore increasing the miR-222 target genes *MyoG*, *MyoD*, and *ER-α*, which are necessary for the differentiation of myoblasts [89]. Dietary daidzein decreased the mRNA and protein expression of ubiquitin-specific protease 19, which is a negative regulator of muscle mass, through the recruitment of ERβ, and increased soleus muscle mass in young female mice but not in male mice [90]. In cisplatin-induced LLC-tumor-bearing mice, daidzein attenuated the loss of myofiber CSA and prevented changes in fiber-type proportion. Moreover, it inhibited protein degradation by suppressing atrogin-1 and MuRF1 via the regulation of the Glut4/AMPK/FoxO pathway [91]. A similar effect was suggested by an in vitro experiment, in which cisplatin-induced C2C12 myotube atrophy was reversed by inhibiting the Glut4/AMPK/FoxO pathway [91]. Furthermore, In Dex-induced atrophy in C2C12 myotubes, daidzein prevented myotube atrophy and promoted myotube growth and myogenic differentiation. The promyogenic activity was mediated by two kinases, Akt and P38MAPK, which in turn activated the key myogenic transcription factor MyoD. Moreover, daidzein encouraged myotube growth via the activation of the Akt/mTOR/S6K pathway [92]. In high-fat diet-induced rats, chronic soy protein consumption resulted in improved muscle function, regardless of muscle mass or fiber cross-section area improvement [93]. Formononetin, which is an isoflavone that is found in *Astragalus membranaceus*, ameliorated chronic kidney disease (CKD)-induced muscle atrophy. In formononetin-treated mice, body weight, the weight of the tibialis anterior and GM, and the CSA of skeletal muscles were significantly improved compared with CKD mice. Moreover, it effectively suppressed MuRF-1, MAFbx, and myostatin expression in TNF-α-induced cells and CKD-mice, whereas the phosphorylation of PI3K, Akt, and Foxo3a was enhanced by formononetin in both in vitro and in vivo models [94]. Glabridin, which is a prenylated isoflavone, inhibited Dex-induced protein degradation in C2C12 myotubes and in the tibialis anterior muscle of mice through the suppression of the ubiquitin ligases MuRF1 and Cbl-b but not atrogin-1. Mechanistically, glabridin inhibited the binding of Dex to its receptor, i.e., the glucocorticoid receptor. Moreover, glabridin prevented the Dex-induced phosphorylation of p38 and FoxO3a, which act as upstream molecules to enhance the expression of ubiquitin ligases [95].

### 3.7. Flavonols

Flavonols are PPs that belong to the flavonoid family. They are found in a variety of fruits and vegetables, such as apples, apricots, beans, broad beans, broccoli, cherry tomatoes, chives, cranberries, kale, leeks, pears, onions, red grapes, sweet cherries, and white currants. They exert antioxidant, anti-inflammatory, cardioprotective, neuroprotective, anticancer, antibacterial, and antiviral effects [47,114]. Among the different compounds of this group, such as kaempferol, quercetin, myricetin, and morin, quercetin is the most-studied compound in the context of muscle atrophy as it acts against obesity, disuse, cachexia, and glucocorticoids.

In obesity-induced muscle atrophy, supplementation with quercetin for 9 weeks prevented the reduction of quadriceps and GM mass and muscle fiber size by attenuating protein degradation through MuRF-1 and Atrogin-1 downregulation. It significantly reduced the expression of macrophage or inflammatory cytokines (TNF-α, IL-6, MCP-1, F4/80, and CD68), as well as inflammatory cytokine receptors such as toll-like receptors (TLR) and 4-1BB in skeletal muscle. The transcripts of inflammatory receptors and their signaling molecules (ERK, p38MAPK, and NF-κB) were significantly reduced together with atrogin-1 and MuRF-1 in cocultured myotubes/macrophages [96]. Thus, obesity-induced muscle atrophy was alleviated by quercetin via the modulation of inflammatory cytokines, their receptors, and their downstream signaling pathways [96]. Quercetin suppressed TNF-α-induced C2C12 myotube atrophy by reducing the ubiquitin ligases atrogin-1 and MuRF-1. The downregulation of ubiquitin ligases was triggered by the inhibition of NF-κB and the activation of Heme Oxygenase 1 (HO-1) in myotubes, as the HO-1 inhibitor ZnPP abolished the inhibitory actions of quercetin. In the same study, quercetin suppressed the muscle atrophy caused by obesity by upregulating HO-1 and inactivating NF-κB, which was lost in Nrf-2 deficient mice. These results suggest that quercetin suppresses TNF-α-induced muscle atrophy under obese conditions via Nrf2-mediated HO-1 induction accompanied by the inactivation of NF-κB [97]. Quercetin also limited body weight, GM and tibialis muscle loss in C26-cancer-associated cachectic mice, with a substantial decrease in atrogin-1 but not in MuRF-1 [102]. Similarly, in an A549-cell-injected tumor mouse model, quercetin increased the antitumor activity of trichostatin A (TSA) and suppressed the TSA-induced loss of GM. It further attenuated the TSA-induced activation of FOXO1, atrogin-1, and MuRF-1. Moreover, oxidative damage and inflammatory cytokines were significantly suppressed by quercetin, and it increased MyHC levels in GM [99]. In C2C12 cells, treatment with Dex (250 µM) decreased cell viability and exerted apoptosis via hydroxyl-free radical generation. Quercetin treatment prevented Dex-induced cell viability and apoptosis by regulating mitochondrial membrane potential (ΔΨm) and reducing ROS production. [100]. Moreover, in Dex-induced muscle atrophy in mice, quercetin prevented the loss of the GM by suppressing myostatin and the expression of the ubiquitin ligases atrogin-1 and MuRF-1. The suppression of myostatin was attributed to the phosphorylation of Akt [101]. In tail suspension-induced disuse muscle atrophy, we found that injection of quercetin into the GM prevented its loss. Additionally, it suppressed protein degradation by attenuating the expression of the ubiquitin ligases atrogin-1 and MuRF-1 by decreasing oxidative stress, as manifested by a reduced TBARS level [102]. We also reported the effects of quercetin in denervation-induced muscle atrophy, in which pre-intake of quercetin prevented the loss of muscle mass and the atrophy of GM fibers by opposing decreased mitochondrial genesis and increased mitochondrial hydrogen peroxide release. Overall, it increased the antioxidant capacity of mitochondria, together with increased PGC-1α expression [103]. Furthermore, quercetin decreased the 3D-clinorotation-induced expression of atrogin-1 and MuRF-1 via the dephosphorylation of ERK [60], which is indicative of its efficacy in disuse muscle atrophy. Interestingly, long-term (24 weeks) oral treatment with quercetin glycoside effectively improved motor performance and increased muscle (quadratus femoris, gastrocnemius, tibialis anterior, and soleus) mass during the early stages of aging [104]. The stimulation of mitochondrial biogenesis by quercetin was also reported in a mouse experiment. Intake of quercetin for 7 days increased the mRNA expression of PGC-1α and SIRT1, mtDNA, and cytochrome c concentration. Both the maximal endurance capacity and voluntary wheel-running activity of mice were enhanced by quercetin supplementation [105].

Morin is another compound in the flavonol group that is commonly found in plants of the Moraceae family [115]. We reported the antiatrophic effect of morin in cachexia and in a Dex-induced muscle atrophy model. In LLC cell-bearing mice, intake of morin prevented the reduction of muscle wet weight and myofiber size by suppressing cancer growth via binding to the ribosomal protein S10 (RPS10) [106]. In Dex-induced muscle atrophy in C2C12 myotubes, morin attenuated E3 ubiquitin ligases atrogin-1, MuRF-1, and Cbl-b and preserved the fast-type and slow-type MyHC protein content. These actions were mediated by a decrease in oxidative stress and an increase in the phosphorylation of Foxo3a. Morin also upregulated PGC-1α [107].

### 3.8. Anthocyanins

Anthocyanins are natural pigments belonging to the flavonoid family. The blue, purple, red, and orange color of many fruits and vegetables stem from the presence of anthocyanins [116]. The most common anthocyanidins in plants are delphinidin, cyanidin, petunidin, pelargonidin, peonidin, and malvidin. Berries, red grapes, cereals, purple maize, vegetables, and red wine are the chief sources of anthocyanins. They exhibit various health benefits, including antioxidant, anti-inflammatory, anticancer, antidiabetic, hepatoprotective, and neuroprotective effects [116,117]. The antiatrophic effects of delphinidin and cyanidin have been reported.

Delphinidin, which is one of the major anthocyanidins, suppresses MuRF1 expression by preventing the downregulation of miR-23a and the nuclear factor of activated T cells (NFATc3) in Dex-induced C2C12 cells [108]. In the same in vivo study using tail-suspended mice, delphinidin suppressed GM loss by reducing MuRF-1 expression and increasing miR-23a and NFATc3. miR23a, which is a micro-RNA (miRNA), decreases MuRF-1 expression by suppressing its mRNA translation. Hence, the beneficial effect of delphinidin occurred via the modulation of the NFATc3/miR-23a pathway [109]. Similarly, delphinidin suppressed the Dex-induced expression of Cbl-b in C2C12 myotubes. In the same study, oral administration of delphinidin attenuated the loss of quadriceps muscle induced by unloading in mice. These effects were mediated by the attenuation of the expression of Cbl-b and stress-related genes [109]. In our previous study, we found that delphinidin insignificantly suppressed the LPS-induced expression of atrogin-1 [82]. A cyanidin-rich diet delayed the progression of muscular dystrophies by attenuating inflammation and oxidative stress in dystrophic alpha-sarcoglycan (Sgca) gene null mice [110].

### 3.9. Stilbene

Stilbenes are an important group of nonflavonoid phytochemicals with a polyphenolic structure. They are found in many plant species, including grapes, peanuts, sorghum, berries, and wine, and exert cardioprotective, chemopreventive, antiobesity, antidiabetic, and neuroprotective properties [118]. Among all identified stilbenes, resveratrol has been studied vigorously because of its wide range of biological activities [119]. It has also been studied in various muscle atrophies induced by immobilization, inflammation, obesity, and glucocorticoids. The effects of stilbene in muscle health have been discussed below and summarized in Table 3.

In a denervation-induced muscle atrophy model, resveratrol prevented the loss of GM mass and fiber atrophy by suppressing the accumulation of the ubiquitin ligases atrogin-1 and p62 in muscle fibers. p62/SQSTM1 is a marker of the functional defect in autophagy–lysosome signaling, which is upregulated during atrophy [120]. In hindlimb-suspended aged rats, resveratrol did not prevent body or muscle weight loss; however, it induced favorable changes in type IIA and IIB muscle fiber CSA and attenuated apoptotic signaling during reloading followed by unloading [121]. Hence, it may be modestly beneficial to patients with sarcopenia, in whom type II fibers are preferentially atrophied. Furthermore, in another disused muscle atrophy model, it preserved soleus muscle mass, maximal force contraction, and mitochondrial capacity. Additionally, it enhanced the protein content of Sirt-1 and COX-IV in the soleus muscle, with suppression of oxidative stress. Primarily, it acted as an exercise mimetic under disuse conditions [122]. In a similar model in old rats, resveratrol found to attenuate atrophy of the GM. It also subdued oxidative stress by enhancing the antioxidant enzymes, catalase, MnSOD, and MnSOD while decreasing the levels of hydrogen peroxide and lipid peroxidation. Moreover, it enhanced the antiapoptotic activity by reducing caspase-9 and increasing Bcl-2 levels [123]. In Dex-induced L6 myotube atrophy, resveratrol inhibited protein degradation and myotube atrophy. Furthermore, it prevented atrogin-1 and MuRF-1 expression induced by Dex by blocking Foxo1 and activating Sirt-1. The protective effect of resveratrol against the Dex-induced effects was mediated by SIRT-1, as deletion of SIRT-1 abolished the effect of resveratrol [124]. In streptozotocin (STZ)-induced diabetic mice, resveratrol attenuated body weight and quadriceps, gastrocnemius, and tibialis anterior muscle weight loss and improved muscle function. It also decreased the expression of the ubiquitin mRNA and MuRF-1 protein, with a simultaneous decrease in LC3-II and cleaved caspase-3. Moreover, resveratrol increased mitochondrial content and mitochondrial biogenesis and inhibited mitophagy in skeletal muscle, as manifested by the increase in succinate dehydrogenase (SDH) activity, Nrf1, COX-IV, PGC-1α, and mtTFA, and the decrease in BNIP3L and phosphorylated Parkin levels, respectively. Furthermore, diabetes-induced fission and fusion of mitochondria were alleviated by resveratrol treatment [125]. Hence, resveratrol prevents diabetes-induced atrophy by reducing the UPS, autophagy, and apoptosis while improving mitochondrial quality and biogenesis. In CKD-induced muscle atrophy in mice, resveratrol prevented the reduction of muscle mass and improved the CSA of the TA muscle by enhancing protein synthesis and reducing protein wasting. Protein synthesis measured by 14C-Phe incorporation in the soleus and EDL muscles was increased in the resveratrol-treated group. Additionally, this drug blocked MuRF-1 expression and the activation of its upstream regulator, NF-κB. Similar results were obtained in Dex-induced C2C12 cells, which were lost in NF-κB-transfected myotubes. Therefore, the effect of resveratrol was mediated by the deactivation of NF-κB, which induces MuRF-1 and protein degradation [126]. In TNF-α-induced atrophy in C2C12 myotubes, resveratrol prevented myotube atrophy by inducing hypertrophy and preventing protein degradation in C2C12 myotubes. Hypertrophy was induced by the phosphorylation of Akt, mTOR, and p70S6K, and 4E-BP1 and proteolysis were attenuated by the downregulation of the ubiquitin ligases atrogin-1 and MuRF-1 via the reduction of the TNF-α-induced Foxo1 levels. Here, the protective effect of resveratrol was mediated by FOXO1, as transfection of FOXO1 eliminated the effect of resveratrol [127]. In cachexia-induced muscle atrophy in mice, resveratrol inhibited skeletal muscle atrophy by inhibiting NF-κB (p65) activity [128]. In sarcopenic obese rats, resveratrol inhibited muscle mass loss and muscle fiber size. It prevented mitochondrial dysfunction and oxidative stress in aged animals with sarcopenia to improve protein metabolism. The same in vitro study reported that resveratrol attenuated the palmitate-acid-induced reductions in MyHC content and myotube diameter in L6 myotubes [129]; however, the studies reported by Jackson et al. in 2011 claimed that resveratrol reduces oxidative stress and preserves fast-twitch fiber contractile function but does not protect against sarcopenic muscle loss [130]. Resveratrol evoked myotube hypertrophy under differentiating media incubation conditions by favoring “slower” MyHC gene expression; moreover, it acutely ameliorated impaired myotube growth during glucose restriction [131].

Resveratrol in combination with other PPs has yielded effective results in the context of muscle health. Resveratrol in combination with astaxanthin and β-carotene enhanced protein synthesis and soleus muscle mass during hypertrophy followed by an immobilization period in mice. Moreover, it increased the phosphorylation of mTOR and p-70S6K while decreasing the carbonylation of proteins [132]. Resveratrol administered together with exercise prevented sarcopenic muscle loss and function in aged rats. It also inhibited apoptosis and improved muscle quality by activating the AMPK/Sirt1 pathway [133]. Similarly, resveratrol together with exercise training attenuated the aging-related mitochondrial dysfunction involving Bad, caspase 3, and IL-6 expression in SAMP8 mice [134]; additionally, it induced hypertrophy in the skeletal muscles of sarcopenic SAMP8 mice. Hence, combined therapy of resveratrol and exercise may attenuate obese sarcopenia. Moreover, treatment with resveratrol and curcumin enhanced the number of satellite cells (muscle progenitor, quiescent, activated, and total satellite cells) in the unloaded limb muscles but not in the reloaded muscles; by contrast, in reloaded muscles, resveratrol improved fast-twitch fibers, sirtuin-1 content, and the counts of progenitor muscle cells [135].

### 3.10. Lignans

Lignans are bioactive compounds with a steroid-analogous chemical structure that are considered as phytoestrogens. They are found at relatively low concentrations in various seeds, grains, fruits, and vegetables, and at higher concentrations in sesame and flax seeds. They exhibit numerous biological activities, including anti-inflammatory, antioxidant, antitumor, and cardioprotective effects [136]. Schisandrin A, a lignan extracted from *Schisandra chinensis* (SC), shows anti-inflammatory, hepatoprotective, and nephroprotective effects [137]. SC is a traditional Chinese herbal medicine in which the main active components are lignans. Different extracts of SC have been reported to alleviate Dex-, denervation-, and disuse-induced muscle atrophy, as summarized in Table 4 and discussed in this section.

Schisandrin A treatment significantly increased TA muscle weight, muscle fiber size, and grip strength by increasing protein synthesis and decreasing protein breakdown in Dex-induced muscle atrophy in mice. It also attenuated the expression of myostatin and the E3 ubiquitin ligases atrogin-1 and MuRF-1 while increasing the expression of MyHC, both in vivo and in vitro. These effects of Schisandrin A were mediated by the regulation of the Akt/FoxO and Akt/70S6K pathways [138]. Similar results were obtained using an ethanolic extract of Fructus Schisandrae (FS), the fruit of SC, in Dex-treated mice. FS significantly prevented the Dex-induced decrease in GM mass, muscle strength, fiber diameter, and serum lactate dehydrogenase levels. Moreover, it decreased the mRNA expression of myostatin, atrogin-1, and MuRF-1 while upregulating the expression of PI3K, Akt1, adenosine A1 receptor (A1R), and transient receptor potential cation channel subfamily V member 4 (TRPV4) which are associated with muscle growth. It also suppressed the oxidative stress and fibrosis induced by Dex in FS-treated mice [139]. The same authors reported the effect of FS in a denervation-induced muscle atrophy model. FS reduced the markers of muscle damage and fibrosis, inflammatory cell infiltration, cytokines, and apoptosis, and increased the markers of muscle mass and activity. Additionally, it suppressed MDA and reactive oxygen species (ROS) content in the GM while increasing the expression of SOD1, catalase, and GSH. Moreover, it upregulated muscle-specific mRNAs involved in muscle protein synthesis, such a PI3K85α, Akt1, A1R, and TRPV4, and downregulated those involved in protein degradation, including atrogin-1, MuRF-1, myostatin, and SIRT1. Denervation-induced IL-1β and TNF-α were effectively attenuated by FS [140]. In summary, the effects of FS described above were suggested by its antioxidant and anti-inflammatory activities, which led to enhanced protein synthesis and decreased protein degradation. FS treatment enhanced myogenic differentiation and muscle hypertrophy through mTOR/P70S6K and 4E-BP1 signaling in human skeletal myotubes [141]. In ovariectomized rats, SC treatment combined with exercise alleviated sarcopenic muscle wasting and enhanced muscle regeneration in skeletal muscle by promoting mitochondrial biogenesis and autophagy. In the same study, which used an in vitro model, SC significantly reduced the expression of inflammatory markers and β-galactosidase activity while improving the antioxidant defenses by decreasing ROS levels [142]. Furthermore, in forced-swimming exercise-induced aged mice, SC effectively increased the strength and thickness of the GM and soleus muscles. Finally, SC synergistically upregulated the expression of mRNAs related to protein synthesis (Akt1 and PI3K) and muscle growth (A1R and TRPV4) and downregulated mRNAs related to protein degradation (atrogin-1 and MuRF1) and muscle growth inhibition (myostatin and SIRT1) [143].

Magnolol is another important compound in the lignan class that is extracted from *Magnolia officinalis* [144]. Magnolol significantly attenuated body weight and muscle weight loss in cisplatin-induced sarcopenic mice. The diameter of the TA muscle was improved by magnolol treatment. Moreover, it increased insulin-like growth factor (IGF-1) expression by enhancing M2c macrophages. In turn, M2c macrophages stimulate myotube formation by expressing high levels of IGF-1 and anti-inflammatory cytokines, such as IL-10 and TGF-β [145]. Moreover, magnolol attenuated myotube atrophy in C2C12 cachectic myotubes by promoting protein synthesis and decreasing ubiquitin-ligase-mediated protein degradation. Myostatin expression and the phosphorylation of SMAD2/3 were decreased, whereas Foxo3a phosphorylation was enhanced via magnolol treatment. Additionally, the expression of MyHC, MyoG, and MyoD was upregulated by the activation of the Akt/mTOR-regulated pathway and decreased the myostatin-mediated expression of atrogin-1 and MuRF-1 [146]. Treatment with magnolol together with chemotherapeutic agents, such as gemcitabine and cisplatin or gemcitabine, significantly suppressed body weight loss and skeletal muscle atrophy compared with conventional chemotherapy. Magnolol inhibited myostatin and activin A, as well as FOXO3a transcriptional activity via Akt activation. Activation of Akt also suppressed MuRF-1, atrogin-1, and proteasomal enzyme activity. Intriguingly, magnolol induced IGF-1 production and related protein synthesis via the PI3K/AKT/mTOR pathway [147]. Sesamin is another important lignan that improves mitochondrial function and exercise capacity by inhibiting NAD(P)H oxidase-dependent oxidative stress in the skeletal muscle of HF-diet-induced diabetic mice [148].

### 3.11. Other Compounds

#### Curcumin

Curcumin, which is a derivative of ferulic acid, is the main natural polyphenol found in the rhizome of *Curcuma longa* (turmeric) and in other *Curcuma* spp. It has been traditionally used in Asian countries as a medicinal herb because of its antioxidant, anti-inflammatory, antimutagenic, antimicrobial, and anticancer properties [149]. It is also extensively used in the Indian subcontinent as a spice in cooking. Curcumin targets multiple signaling molecules, thus exhibiting a wide range of health benefits. Various studies have reported that curcumin effectively attenuates disuse- and inflammation-induced muscle atrophy in multiple in vitro and in vivo models, which are summarized in Table 5 and discussed below.

In LPS-induced muscle atrophy in mice, curcumin suppressed the loss of GM mass and protein. This effect was mediated by a reduction in the expression of the ubiquitin ligases atrogin-1, together with the inhibition of p38 MAPK. However, curcumin was ineffective in suppressing the expression of MuRF-1 and its upstream regulator, NF-κB [150]. Moreover, in COPD-induced muscle atrophy in rats, curcumin attenuated muscle fiber atrophy, myofibril disorganization, and mitochondrial damage. The activities of cytochrome *c* oxidase, succinate dehydrogenase, Na^+^/K^+^-ATPase, and Ca^2+^-ATPase were significantly improved in the mitochondria of skeletal muscles from COPD rats. Additionally, it decreased oxidative stress and inflammation (IL-6 and TNF-α) while increasing the mRNA and protein expression of PGC-1α and SIRT3 in skeletal muscle tissues, which may be attributed to its improved mitochondrial function [151]. Furthermore, in STZ-induced type-I diabetic mice, curcumin attenuated muscle atrophy by inhibiting protein ubiquitination but not protein synthesis. It significantly decreased ubiquitin-conjugated proteins and the mRNA expression of atrogin-1 and MuRF-1, with the prevention of the activation of NF-κB. These effects were detected because of the inhibition of inflammatory cytokines (TNF-α and IL-β) and oxidative stress by curcumin [152]. In CKD-induced muscle atrophy in mice, curcumin treatment preserved body weight and improved muscle functions. It also enhanced mitochondrial biogenesis and mitochondrial function by promoting ATP levels as well as the activities of the mitochondrial electron transport chain, with the suppression of mitochondrial oxidative stress. The protective effects described above were mediated by the inhibition of glycogen synthase kinase 3 beta (GSK-3β) activity in vitro and in vivo, as the knockout of GSK-3β could not improve mitochondrial function and quality [153]. Under cachexia conditions in MAC16 colon tumor-bearing mice, supplementation with the curcumin C3 complex prevented the loss of body weight, GM weight, and fiber CSA. Molecularly, it reduced proteasome complex activity and the expression of atrogin-1 and MuRF-1, to attenuate protein degradation [154]. Similarly, under hypobaric hypoxia-induced muscle atrophy in mice, curcumin attenuated muscle atrophy and protein degradation with increased myofibrillar proliferation and differentiation. Additionally, curcumin prominently reduced hypoxia-induced oxidative stress, which causes protein degradation via calpain and the ubiquitin–proteasome pathway [155]. In disuse-induced muscle atrophy in vivo, curcumin prevented the loss of soleus mass and myofiber CSA by decreasing oxidative stress and upregulating Grp94 levels [156]. Another study found that curcumin improved the recovery of muscle mass and fiber CSA during reloading by blunting proteasome chymotrypsin-like activity and apoptosome activity [157]. Treatment with curcumin triggered sirtuin-1 activity while suppressing proteolysis in the GM of mice during reloading followed by unloading. Muscle proteolysis was attenuated possibly via the activation of the histone deacetylase sirtuin-1, which is attributed to a decrease in the levels of atrophy signaling pathways [158]. Curcumin in combination with fish oil mitigated the unloading-induced decrease in myofiber CSA. It also upregulated HSP70 and anabolic signaling (phosphorylation of Akt and p70S6K) while reducing oxidative stress. In summary, treatment with curcumin together with fish oil prevents unloading-induced atrophy by upregulating HSP70 and the anabolic pathway [159].

## 4. Toxicity of Polyphenols

Various in vitro and in vivo toxicological studies suggest that polyphenols are safe phytochemical agents. However, they may induce toxic effects depending on dose and duration of treatment. The experimental data depicting toxicity of polyphenols has been shown in Table 6.

## 5. Critical Overview and Future Perspectives

This review has summarized the effects of different polyphenolic compounds in improving muscle health and alleviating the muscle disorders induced by various proatrophic factors. The results of PPs administration in multiple muscle atrophy models in vitro and in vivo were presented together with their possible mechanisms of action. According to the results reviewed above, polyphenolic compounds stimulate protein synthesis and impede protein degradation to attenuate muscle atrophy by mediating different anabolic and catabolic pathways. PPs such as FA, EC, 8-PN, daidzein, resveratrol, and Schisandrin A effectively activated the Akt/mTOR pathway of protein synthesis [41,54,77,92,128,139]. However, the majority of the polyphenolic compounds exerted their antiatrophic effect by preventing or reducing protein degradation in atrophic muscles. Mainly, the E3 ubiquitin ligases atrogin-1, MuRF-1, and/Cbl-b were suppressed by PPs in muscle atrophies caused by inflammation, oxidative stress, and mitochondrial damage. The upstream regulators of E3 ubiquitin ligases, i.e., TNF-α, IL-6, IL-1β, NF-κB, myostatin, and FoxOs, were significantly suppressed by PPs [32,38,79,96,127,142,151]. Moreover, PPs modulated the expression of miRNAs associated with muscle atrophy. miR-486-5p, miR-23a, and miR-222 have been reported to have an effect on muscle health [56,84,115]. Upregulation of miRNA-486-5p by EGCG effectively decreased Foxo1-mediated ubiquitin ligase expression [62]. Similarly, miR-23a expression was enhanced by delphinidin [108], which prevented the translation of MuRF-1. Likewise, the expression of miR-222 was downregulated by AP, which enhanced the expression of the miR-222 target genes MyoG, MyoD, and ER-α, which are required for myoblast differentiation [89]. Hence, both catabolic/anabolic signaling molecules and miRNAs are the target of PPs for their antiatrophic effects (Figure 2).

Generally, in muscle health-related research, compounds that are capable of modulating catabolic and anabolic pathways are studied with higher priority. However, for proper muscle functioning and activity, mitochondrial quality and myogenesis are indispensable factors that should also be considered. Mitochondria not only produce cellular energy but also regulate the antioxidant and apoptosis phenomena. In this review, we reported several compounds that enhance mitochondrial functions by increasing mitochondrial biogenesis or reducing mitochondrial dysfunction [29,36,56,74,78,79,105,125]. Furthermore, several polyphenolic compounds were found to stimulate myogenesis for the growth and differentiation of muscle cells [30,71,81,146]. Thus, for future antiatrophic treatment, both the catabolic/anabolic modulators and mitochondrial bioenergetics will be useful therapeutic targets. We believe that our report will provide a gross notion of the molecular mechanisms of the action of PPs against muscle disorders.

## 6. Natural Polyphenols Used in Clinical Trials to Treat Muscle Disorders

Although various polyphenolic compounds have been found to be effective against muscle atrophies in a wide range of in vitro and in vivo experiments, clinical trials of polyphenols are scarce due to ambiguities of polyphenols doses, adverse effects, lack of evidence for efficacy and bioavailability in human subjects. The clinical trials of polyphenols to other diseases have been proposed [26], but they are still to be trialed in muscle atrophies due to above reasons. Interestingly, a study reported that the administration of epicatechin enriched tannase-treated green tea extract for 12 weeks significantly increased muscle, grip strength and muscle mass in individuals aged 60 years or older [179]. Furthermore, myostatin, a negative regulator of muscle mass, was reduced following administration of tannase-treated green tea extract. Moreover, Ramirez-Sanchez et al. (2013) reported that treatment of epicatechin rich cocoa (ERC) to type-2 diabetes (T2D) and heart failure (HF) patients prevents alternations in oxidative stress regulatory system. It induced the expression of key antioxidants SOD2, catalase and glutathione in the skeletal muscle of patients. Additionally, ERC decreased the nitrotyrosilation and carbonylation of skeletal muscle protein [180]. Similarly, Taub, Pam R et al. (2012) found that administration of ERC to the T2D/HF patients improves mitochondrial structure and markers of mitochondrial biogenesis [181]. Treatment of ERC also enhanced dystrophin-associated protein complex (DAPC) protein level, sarcomeric microstructure and markers of skeletal muscle growth/differentiation [182] in clinical application.

Therefore, we believe that published studies on polyphenols against muscle atrophies could be particularly helpful to evaluate the safety, efficacy, and actual dose determination for future clinical application, however further investigations are required for optimizing the clinical use of polyphenols against muscle atrophy.

## 7. Current Therapeutic Strategies Implemented Targeting Skeletal Muscle Atrophy

Several therapeutic strategies have been tried and proved to have beneficial effects in the treatment of muscle atrophy. However, most of the studies were carried out in in vitro and in vivo models, lacking their effects in human subjects. Presently, exercise is the only proven treatment to attenuate muscle atrophy, which is not convenient for severely ill, bedridden, and neurologically impaired patients. As muscle atrophy is associated with multifactorial pathogenesis, a combination of nutraceutical, synthetic drugs, and exercises may be the most effective strategy to prevent muscle atrophy. Here, we discussed some potential drugs under laboratory or preclinical research to attenuate muscle atrophy. 

### 7.1. Anabolic Medications

a.Androgen (testosterone) or androgen receptor modulator (Enobosarm, GTx-024):These drugs can induce muscle protein synthesis by activating IGF-1 signaling. Although testosterone improves muscle mass, its consumption is discouraged due to its side effects such as prostrate hypertrophy, cancer, sleep apnea, and thrombosis [183,184]. Androgen receptor modulators show similar effects as testosterone; however, they show lesser adverse effects compared to testosterone [185]. b.Ghrelin and its receptor agonist (anamorelin):Ghrelin is a growth hormone releasing polypeptide. Numerous studies show that Ghrelin reduces muscle atrophy induced by various pro-atrophic factors via reducing inflammatory cytokines and their downstream molecules, along with restoring expression of p-Akt and p-FoxO1. In a clinical study, both ghrelin and its agonist anamorelin showed increased lean body mass and muscle strength in cachexia patients. Anamorelin has a higher half-life and oral route of administration whereas Ghrelin’s half-life is only 0.5 hr with an intravenous route of administration [186,187].c.β2-Adrenoceptor agonists (Formoterol, Clenbuterol, espindolol):These drugs may regulate muscle mass by activating G-protein coupled β-adrenoceptor, that activates protein kinase A and hence stimulate PI3K/Akt/mTOR signaling. They also increase level of follistatin and decrease myostatin level. Due to their cardiovascular side effects, they are less preferentially applied in clinical application [188,189]. 

### 7.2. Enzyme Inhibitors

a.Cox2 Inhibitors (Celecoxib, Meloxicam): Cox2 and its downstream molecule prostaglandin (PGE2) regulate the cytokines activity and mediate cachexia. These drugs found to suppress cachexia and inflammatory cytokine in clinical study [190].b.Histone deacetylase Inhibitors (Trichostatin): Drugs of this group regulate atrogenes expression and muscle mass by reducing HDAC4 activity. By inhibiting HDAC activity, Trichostatin inactivates FoxOs that mediates muscle atrophy [191].c.PDE inhibitors (Torbafyllin, rolipram, cilomilast): These drugs act as inhibitor of phosphodiesterase (PDE), which stimulate proteolysis [192].d.ACE- inhibitors (enalapril, Perindopril): These drugs prevent muscle atrophy by inhibiting conversion of Angiotensin-I to angiotensin-II, which suppress protein anabolism and promote protein catabolism increasing ROS, inflammatory cytokines, and glucocorticoids in skeletal muscle [193].

### 7.3. Anti-Inflammatory Drugs

Thalidomide, anti-IL-6/STAT3 (Tocilizumab), anti-TNF-a(Infliximab), anti-IL-1a (MABp1), etc. can prevent muscle atrophy by suppression the inflammatory cytokines and their downstream effectors [194].

### 7.4. Natural Compounds

Various natural products such as quercetin, resveratrol, salidroside, and isoflavones can prevent muscle atrophy by suppressing inflammation, oxidative stress, and catabolic pathways while stimulating the anabolic signaling [194]. However, more studies are warranted to use them in clinical application.

## 8. Conclusions

The polyphenols reported in this review have been found to prevent muscle atrophy via the suppression of muscle protein degradation, mainly by downregulating ubiquitin ligases, and the promotion of protein synthesis, myogenesis, and mitochondrial quality in in vitro and in vivo studies, as shown in Figure 3. The main mechanisms associated with the beneficial effects of PPs on muscle atrophy are: inhibition of inflammatory cytokines, oxidative stress, myostatin, and muscle atrophy-related ubiquitin ligases: Atrogin-1, MuRF-1, and Cbl-b. Anti-atrophic effects of PPs are also mediated by activation of the IGF-1 signaling pathway, follistatin, miRNAs, mitochondrial biogenesis, and myogenic differentiation factors involved in myogenesis. Therefore, PPs having effects on the attenuation of muscle proteolysis and maintenance of muscle mass could facilitate the avenue for designing suitable therapeutic strategies for the attenuation of skeletal muscle atrophy and the promotion of muscle health.

Overall, this review has outlined the available information on the effects of PPs on muscle health in cellular and rodent models, which could guide investigators in the study of PPs more extensively, to establish their use as therapeutic agents against muscle atrophy. It also shed light on the effects of PPs on mitochondrial quality control and muscle myogenesis. However, issues such as safety, bioavailability, exact dose determination, the metabolites of polyphenolic compounds, and their real molecular mechanisms obstruct the use of PPs as therapeutic drugs for muscle atrophy treatment. Therefore, further studies are required to address these issues and promote the translation of the positive effects of PPs into clinical outcomes.

## Figures and Tables

**Figure 1 molecules-26-04887-f001:**
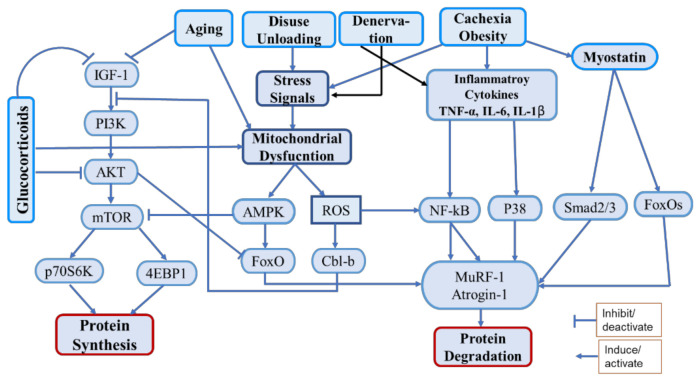
Schematic diagram of the main pathways associated with protein synthesis and degradation under various proatrophic condition.

**Figure 2 molecules-26-04887-f002:**
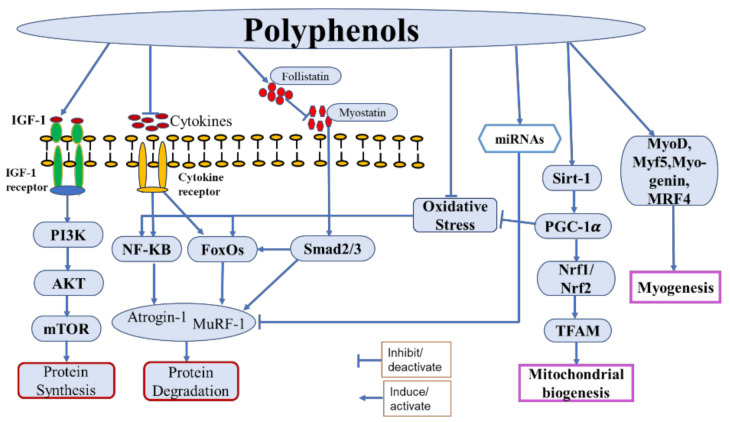
Schematic illustration of the possible representative pathways of polyphenols action in the treatment of skeletal muscle disorders and improvement of muscle health.

**Figure 3 molecules-26-04887-f003:**
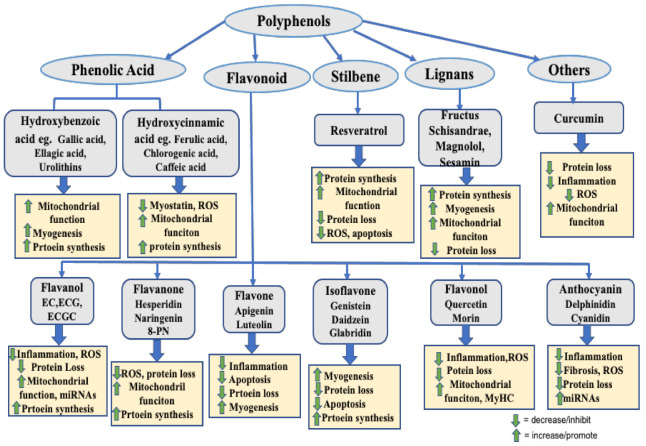
Schematic illustration showing members of polyphenols representative with impact on skeletal muscle.

**Table 1 molecules-26-04887-t001:** Function of phenolic acids and their derivatives in promotion of muscle health and prevention of muscle atrophy. DMD, Duchenne muscle dystrophy; EC, epicatechin; EGC, epigallocatechin; GA, gallic acid; HF: high fat diet.

Class	Sub-Class	Compound/Derivatives/Compounds Mixture	Model	Effects	References
**Phenolic Acid**	**Hydroxybenzoic Acid**	Gallic Acid (GA)	C2C12 Myotubes	Increased Mitochondrial Function and mitochondrial biogenesis, Enhanced myosin heavy chain expression	[29]
		EC, EGC and GA	Normal and oxidative stress-induced C2C12 cells	Increased myotube density upregulated genetic expression of myogenic factors	[30]
		Ellagic acid	CCL4-induced muscle injury in rats	Reduced muscle tissue damage induced caspase-3, Nrf-2 and antioxidant enzymes suppressed inflammatory markers	[32]
		Ellagic Acid	Cuprizone-induced multiple sclerosis model in mice	Protects muscle tissue prevented mitochondrial dysfunction and oxidative stress	[33]
		Urolithin A	C2C12 cells, young and HF-induced aged mice	Induced autophagy and mitophagy both invitro and in vivo increased muscle function improved exercise capacity	[34]
		Urolithin A	Mouse model of DMD	Induced mitophagy and improved muscle function and MuSCs regeneration increased skeletal muscle respiratory capacity	[35]
		Urolithin A	C57BL/6 mice	Strengthen skeletal muscle and angiogenesis Increase ATP and NAD+ level Upregulates angiogenic pathways	[36]
		Urolithin B	C2C12 myotubes and denervation-induced mice	Enhanced growth and differentiation of C2C12 myotubes and muscle hypertrophy Increased protein synthesis and suppressed UPS	[37]
		Pomegranate extract	TNF-α induced muscle atrophy in mice	Prevented muscle wasting suppressed cytokines and NF-kB level activated protein synthesis pathway	[38]
	**Hydroxycinnamic Acid**	Ferulic acid	Mouse C2C12 myotubes	Regulates muscle fiber type formation activated SIRT1/AMPK pathway Increased PGC-1α expression	[39]
		Ferulic Acid	Corticosteroid-Induced Rat Myopathy	Induced growth of fast glycolytic and slow oxidative muscle fiber suppressed myostatin and oxidative stress	[40]
		Ferulic Acid	Zebrafish model	Enhanced muscle mass and MyHC fast type Increased myogenic transcriptional factors activated zTOR/p70S6K/4EBP1	[41]
		Chlorogenic acid	Resistance training-induced rat model	improved muscle strength by promoting mitochondrial function and cellular energy metabolism	[42]
		Caffeic acid phenethyl ester	Eccentric exercise-induced skeletal muscle injury in rats	Protected skeletal muscle damage down-regulated NF-κB activation	[43]
		Caffeic acid	Human fibroblast cell line	Decreased spinal muscular atrophy increased SMN2 transcripts	[44]
		Coffee	In-vitro and in-vivo model	Skeletal muscle hypertrophy and myoblast differentiation	[45]
		P-Coumaric acid	C2C12 myotubes	Reduce differentiation of muscle cells by reducing MyoD and Myogenin.	[46]

**Table 2 molecules-26-04887-t002:** Function of flavonoids and its derivatives in promotion of muscle health and prevention of muscle atrophy. 8-PN, 8-prenylnaringenin; Dex, dexamethasone; EGCG, epigallocatechin gallate.

Class	Sub-Class	Compound/Derivatives	Model	Effects	References
**Flavonoids**	**Flavanols**	EC	Young and old C57BL/6 mice and human tissue samples	Decreased myostatin and β-galactosidase Increased follistatin and markers of muscle growth and differentiation	[52]
		EC	Young and old C57BL/6 mice	Increased survival of aged mice prevented muscle wasting	[53]
		EC	HLS-induced muscle atrophy in mice	Counteracts muscle degradation maintains muscle angiogenesis and mitochondrial biogenesis	[54]
		EC	Denervation-induced muscle atrophy in rats	Reduced muscle wasting down regulated Foxo1a, Atrgoin-1 and MuRF-1	[55]
		EC	Ambulatory adults with Becker Muscular Dystrophy	Induced mitochondrial biogenesis and muscle regeneration factors increased follistatin decreased myostatin	[56]
		EC	C2C12 myotubes	Stimulates mitochondrial biogenesis and cell growth through GPER	[57]
		EC	C2C12 cells	Enhanced myogenic differentiation and MyoD activity	[58]
		EC	HF-induced muscle damage in aged mice	Improved physical performance increased follistatin and MEF2A decreased Foxo1a and MuRF-1	[59]
		EC, ECG, EGCG	3D-clinorotation induced C2C12 atrophy	Suppressed Atrogin-1 and MuRF-1 dephosphorylated the ERK signaling	[60]
		EGCG	Aging-induced sarcopenic rat model	Attenuated muscle atrophy and protein degradation Increased protein synthesis	[61]
		EGCG	Aged mice and late passaged C2C12 cells	Attenuated muscle atrophy and protein degradation upregulated miR-486-5p	[62]
		EGCG	Starvation and TNF-α-induced C2C12 atrophy	Attenuation of muscle wasting inhibited protein degradation and activated protein synthesis	[63]
		EGCG	Tumor induced LLC and C57Bl/6 mice atrophy	Attenuates muscle atrophy inhibits NF-κB, atrogin-1 and MuRF-1	[64]
		EGCG	Mouse model of Duchenne muscular dystrophy	Protected from muscle necrosis improved muscle functions	[65]
		EGCG	HLS-induced muscle atrophy in aged rats	Improved Plantaris muscle weight and fiber size reduced pro-apoptotic signaling	[66]
		EGCG	HLS-induced muscle atrophy in aged rats	Maintained autophagy signaling in disuse muscle prevented autophagy and apoptosis during reloading	[67]
		EGCG	Dex-induced C2C12 myotubes and tail-suspended mice	Prevented muscle atrophy suppressed MuRF1	[68]
		Green tea extracts (EGCG)	HF diet-induced muscle atrophy in SAMP8 mice	Ameliorated HF-induced muscle wasting Decreased insulin resistance and LECT2 expression	[69]
		EGCG	Live skeletal muscle fibers model	Promoted nuclear efflux of Foxo1 activated PI3K/Akt pathway	[70]
		Catechin (ECG+EGCG)	C2C12 cells and Cardiotoxin induced C56BL/6 mice	Stimulated muscle stem cell activation and differentiation for muscle regeneration.	[71]
		Catechin	Downhill running-induced muscle damage in ICR mice	Attenuated downhill running-induced muscle damage suppressed oxidative stress and inflammation	[72]
		catechins	Tail-suspension induced muscle atrophy in mice	Minimized contractile dysfunction and muscle atrophy decreased oxidative stress	[73]
	**Flavanones**	Hesperidin	Human skeletal muscle cell and mice	Reverted aging-induced decrease in muscle fiber size increased mitochondrial function and running performance reduced oxidative stress	[74]
		Naringenin	L6 and C2C12 cells	Delays skeletal muscle differentiation	[75]
		8-Prenylnaringe-nin (8-PN)	Denervated Mice	Prevented muscle atrophy suppressed Atrogin-1 Phosphorylated Akt	[76]
		8-PN	C2C12 myotubes and casting-induced muscle atrophy in mice	Reversed casting-induced loss of tibialis anterior muscle activated PI3K/Akt/mTOR pathway	[77]
	**Flavones**	Apigenin	Aged Mice	Relieved muscle atrophy and increased myofiber size inhibited hyperactive mitophagy/ autophagy and apoptosis.	[78]
		Apigenin	Obesity-induced muscle atrophy in Mice	Attenuated muscle atrophy and mitochondrial dysfunctions.	[79]
		Apigenin	Denervated-induced muscle atrophy in mice	Protected muscle loss Upregulated MyHC reduced TNF-α and MuRF-1 level	[80]
		Apigenin	C57BL/6 mice and C2C12 cells	Promotes skeletal muscle hypertrophy enhanced myogenic differentiation upregulated Prmt7-PGC-1α-GPR56 pathway	[81]
		Flavones	LPS-induced muscle atrophy in C2C12 myotube	Prevented myotube atrophy suppressed Atrgoin-1 inhibited JNK phosphorylation	[82]
		Luteolin	LLC-induced muscle atrophy in mice	Prevented muscle atrophy downregulated MuRF-1, Atrgoin-1, cytokines/inflammatory markers	[83]
		Luteolin	Dex-Induced muscle atrophy in mice	Prevented muscle atrophy Induced antioxidant and antiapoptotic activity	[84]
	**Isoflavones**	Isoflavone (genistein and daidzein)	TNF-α induced C2C12 myotubes	Suppressed MuRF1 promoter activity and myotube atrophy	[85]
		Isoflavone	Tumor-induced muscle atrophy in mice	Prevented muscle wasting suppressed ubiquitin ligases expression	[86].
		Isoflavone	Denervation-induced muscle atrophy in mice	Prevented muscle fiber atrophy decreased apoptosis-dependent signaling.	[87]
		Genistein	Denervation-induced muscle atrophy in mice	Mitigated soleus muscle atrophy	[88]
		Genistein	C2C12 myoblast	Enhanced proliferation and differentiation downregulated miR-222	[89]
		Daidzein	Young female mice	Down-regulated ubiquitin-specific protease 19 increased soleus muscle mass	[90]
		Daidzein	Cisplatin-induced LLC bearing mice	Alleviated skeletal muscle atrophy prevented protein degradation	[91]
		Daidzein	C2C12 cells	Promotes myogenic differentiation and myotube hypertrophy	[92]
		Soy protein	Rats	Improved muscle function	[93]
		Formononetin (isoflavone)	CKD rats and TNF-α-induced C2C12 myotubes atrophy	Prevented muscle wasting suppressed MuRF-1, MAFbx and myostatin expression Phosphorylated PI3K, Akt and FoxO3a	[94]
		Glabridin	Dex-induced atrophy (in vitro and in vivo)	Inhibited protein degradation and muscle atrophy in vitro and in vivo	[95]
	**Flavonols**	Quercetin	HF-diet induced muscle atrophy in mice	Protected muscle mass and muscle fiber size reduced ubiquitin ligases and inflammatory cytokines	[96]
		Quercetin	TNF-α induced myotube and Obesity induced mice muscle atrophy	Averted muscle atrophy Upregulated HO-1 and Nrf2 Inactivates NF-kB	[97]
		Quercetin	Murine C26 cancer-cachexia model	Prevented body and muscle weight loss tended to decrease Atrgoin-1 and MuRF-1	[98]
		Quercetin	A549 cells injected tumor model in mice	Prevented loss of GM and protein degradation Increased MyHC level	[99]
		Quercetin	Dex-Induced C2C12 cell injury	Increased C2C12 cell viability exerted antiapoptotic effects reduce oxidative stress regulates mitochondrial membrane potential	[100]
		Quercetin	Dex-induced-muscle atrophy in mice	Prevented muscle loss reduced myostatin, atrgoin-1 and MuRF-1 increased Akt phosphorylation	[101]
		Quercetin	Tail-suspension induced muscle atrophy in mice	Prevented GM loss suppressed ubiquitin ligases and lipid peroxidation	[102]
		Quercetin	Denervation-induced muscle atrophy in mice	Prevented muscle atrophy suppressed mitochondrial hydrogen peroxide generation elevated mitochondrial biogenesis	[103]
		Quercetin Glycosides	Male C57BL/6J aged mice	Improved motor performance Increased muscle mass	[104]
		Quercetin	Mice	Mitochondrial biogenesis Increased endurance and running capacity	[105]
		Morin	β cell-bearing mice and C2C12 myotubes atrophy	Suppressed muscle wasting and myofiber size reduction by binding to ribosomal protein S10	[106]
		Morin	Dex- induced atrophy of C2C12 skeletal myotubes	Prevented protein degradation reduced oxidative stress, Atrogin-1, MuRF-1 and Cbl-b Phosphorylated Foxo3a	[107]
	**Anthocyanins**	Delphinidin	Dex-induced C2C12 atrophy and tail-suspension induced atrophy in mice	Suppressed MuRF-1 expression Prevented muscle weight loss upregulated miR-23a and NFATc3	[108]
		Delphinidin	Dex-induced C2C12 atrophy and tail-suspension induced atrophy in mice	Suppressed disused-muscle loss suppressed Cbl-b and stress related gene	[109]
		Delphinidin	LPS-induced atrophy in C2C12 myotubes	Reduced atrogin-1 expression insignificantly	[82]
		Cyanidin	Dystrophic alpha-sarcoglyan (Sgca) null mice	Reduced progression of muscular dystrophy Reduced inflammation and fibrosis	[110]

**Table 3 molecules-26-04887-t003:** Function of stilbene and its derivatives in promotion of muscle health and prevention of muscle atrophy.

Class	Sub-Class	Compound/Derivatives	Model	Effects	References
**Polyphenol**	**Stilbene**	Resveratrol	Denervation-induced muscle atrophy	Prevented loss of muscle weight and fiber CSA reduced atrogin-1 and P62 level	[120]
		Resveratrol	HLS-induced muscle atrophy in aged rat	Improved type IIA and IIB muscle fiber CSA decreased pro-apoptotic protein	[121]
		Resveratrol	HLS-induce muscle atrophy in rat	Prevented soleus muscle loss improved mitochondrial capacity and Sirt-1 and COXIV protein	[122]
		Resveratrol	HLS-induced muscle atrophy in young and aged rats	Prevented GM loss decreased oxidative stress and increased antioxidant defense	[123]
		Resveratrol	Dex-induced L6 myotube atrophy	Prevented myotube atrophy suppressed atrogin-1 and MuRF-1	[124]
		Resveratrol	STZ-induced muscle atrophy in diabetic mice	Prevented muscle atrophy Preserved body weight, muscle mass, muscle function and mitochondrial quality	[125]
		Resveratrol	C2C12 cells and CKD-induced muscle atrophy model in mice	Attenuated muscle atrophy Increased protein synthesis decreased protein degradation	[126]
		Resveratrol	TNF-α-induced muscle atrophy in C2C12 myotubes	Prevented myotube atrophy reduced Foxo1, atrogin-1, MuRF-1 regulates Akt/mTOR/FoxO1 signaling	[127]
		Resveratrol	C26 adenocarcinoma tumors-induced muscle atrophy in mice	Prevented muscle atrophy attenuated NF-kB activity	[128]
		Resveratrol	HF-diet induced obese sarcopenia in aged rat	Increased mitochondrial function Prevented loss of muscle and mitochondrial function	[129]
		Resveratrol	Young (6 months) and middle-aged (18 months) mice	Alleviated oxidative stress preserved fast twitch contractile function	[130]
		Resveratrol	Glucose restriction-induced atrophy	Evoked myotube hypertrophy favored slow type MyHC gene expression	[131]
		Resveratrol astaxanthin and β-carotene	Immobilization-induced muscle atrophy in mice	Prevented soleus muscle loss increased phosphorylation of mTORC1 and p70S6K decreased protein carbonylation	[132]
		Exercise, resveratrol	6-month and 24-month old rats as young and aged model	Increased grip strength and muscle mass in aged rats reduced apoptotic markers	[133]
		Resveratrol + exercise	HF-induced obese sarcopenic SAMP8 mice	Attenuated sarcopenia -related mitochondrial dysfunction	[134]
		Resveratrol and Curcumin	HLS-induced muscle atrophy in mice	Enhanced satellite cells number protected CSA of muscle fibers increased sirtuin-1 activity	[135]

**Table 4 molecules-26-04887-t004:** Function of lignan and its derivatives in promotion of muscle health and prevention of muscle atrophy.

Class	Sub-Class	Compound/Derivatives/Compounds Mixture	Model	Effects	References
**Polyphenol**	**Lignan**	Schisandrin A	Dex-induced C2C12 and C57BL/6 mice muscle atrophy	Prevented muscle atrophy suppressed protein degradation and enhanced protein synthesis	[138]
		Ethanol extract of Fructus Schisandrae	Dex-induced muscle atrophy in mice	Prevented muscle atrophy and protein degradation increased protein synthesis	[139]
		Ethanol extract of Schisandrae Fructus	Denervation-induced muscle atrophy in mice	Attenuated muscle atrophy Increased protein synthesis decreased protein breakdown	[140]
		Schisandrae fructus	Human skeletal muscle cells	Inhibited muscle atrophy Enhanced muscle differentiation and protein synthesis	[141]
		Schisandra chinensis	C2C12 myoblasts and ovariectomized rats	Improved muscle regeneration and mitochondrial biogenesis exhibited anti-inflammatory and antioxidant effects	[142]
		Schisandrae Fructus extract	Chronic forced exercise-induced mice	Enhanced muscle strength increased muscle protein synthesis and protein degradation	[143]
		Magnolol	Cisplatin-induced sarcopenic mice	Attenuated body and muscle weight loss increased IGF-1 expression	[145]
		Magnolol	Cachexia-induced C2C12 myotube atrophy	Inhibited myotube atrophy increased protein synthesis and MyHC, MyoD, MyoG	[146]
		Magnolol	Cachectic mice undergoing chemotherapy	Attenuated muscle atrophy Increased IGF-1-mediated protein synthesis	[147]
		Sesamin	HF-induced diabetic mice	Improved mitochondrial function and exercise capacity reduced oxidative stress	[148]

**Table 5 molecules-26-04887-t005:** Function of other polyphenols and in promotion of muscle health and prevention of muscle atrophy.

	Compounds/Derivatives	Model	Effects	References
**Others**	Curcumin	LPS—induced muscle atrophy in mice	Prevented muscle loss inhibited atrogin-1 expression and P38 activation	[150]
	Curcumin	COPD-induced rat model	Attenuated muscle fiber atrophy improved mitochondrial structure decreased oxidative stress and inflammation	[151]
	Curcumin	STZ-induced diabetic mice model	Prevented skeletal muscle atrophy Inhibited ubiquitin ligases expression	[152]
	Curcumin	CKD-induced muscle atrophy in mice	Protected body weight and fiber CSA Increased mitochondrial biogenesis Suppressed oxidative stress and GSK-3β expression	[153]
	Curcumin c3 complex	Human skeletal myoblast cell and tumor-induced muscle wasting in vivo	Prevented cachexia-induced muscle wasting Decreased 20S proteasome activity Improved muscle characteristics	[154]
	Curcumin	Hypoxia induced muscle atrophy of rat	Reduced muscle protein degradation Increased myofibrillar proliferation and differentiation reduced oxidative stress	[155]
	Curcumin	Tail suspension-induced muscle atrophy in rat	Prevented loss of muscle mass promotes Grp94 protein expression	[156]
	Curcumin	Immobilization-induced muscle atrophy in rats	Improved recovery during reloading prevented proteasome chymotrypsin-like activity and caspase-9-associated apoptosome activity	[157].
	Curcumin	HLS-induced muscle atrophy in mice	Attenuated muscle proteolysis elicited Sirt1 activity	[158]
	Curcumin with fish oil	HLS-induced atrophy in mice	Mitigated unloading-induced decrease in fiber CSA Increased HSP70 level phosphorylated Akt and p70S6K Decreased Nox2 level	[159]

**Table 6 molecules-26-04887-t006:** Experimental data showing toxicity of Polyphenols.

Compound	Dose	Model	Toxicity
Gallic acid	128 mg/kg b.w.	Rats	Not observed [160]
Ellagic acid	3011 mg/kg b.w.	Rats	Not observed [161]
Catechin	764mg/kg b.w.	Rats	Not observed [162]
Ferulic acid	2445 mg kg/b.w.	Rats	LD50 value [163]
Hesperidin	4837.5 mg/kg b.w.	Rats	LD50 value [164]
Naringenin	1250 mg/kg b.w.	Rats	Not observed [165]
Apigenin	50 mg/kg b.w.	Mice	Not observed [166]
Luteolin	5000 mg/kg b.w.	Rats	LD50 value [167]
Genistein	50 mg/k kg b.w.	Rats	Not observed [168]
Daidzein	5000mg/kg b.w.	Mice and Rats	Not observed [169]
Glabridin	400mg/kg b.w.	Mice	Not observed [170]
Quercetin	200–500 mg/kg b.w.	Mice and Rats	Not observed [171]
Morin	356 mg/kg b.w.	Rats	Not observed [172]
Anthocyanin	125–500mg/kg b.w.	Rats	Not observed [173]
Resveratrol	750mg/kg b.w.	Rats	Not observed [174]
Magnolol	240 mg/kg b.w.	Rats	Not observed [175]
Sesamin	280 mg/kg b.w.	Rats	Not observed [176]
Curcumin	5000 mg/kg b.w.	Rats	Not observed [177]
P-coumaric acid	2850 mg kg/b.w.	Mice	LD50 value [178]

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
