# Peer review of "Polyphenols and Their Effects on Muscle Atrophy and Muscle Health"

_molecules, 2021, doi:10.3390/molecules26164887_

Round 1

Reviewer 1 Report

The paper titled with “Polyphenols and Their Effects on Muscle Atrophy and Muscle Health by Takeshi Nikawa et al describes polyphenol actions in skeletal muscle atrophy prevention and muscle health. This is an interesting study and the authors have collected a dataset after a depth search on Pub Med and Google Scholar. Sufficient informations on the results of previous studies are presented to readers to get a general idea of the effect of polyphenols on muscle atrophy. The paper is generally well written and structured.

General comments as follows

  • The authors missed to cite a recent review on the same topic (Salucci, S.; Falcieri, E.; Polyphenols and their potential role in preventing skeletal muscle atrophy. Nutrition Research, 2020, 74, 10-22 https://doi.org/10.1016/j.nutres.2019.11.004
  • Some references (e.g. Ref 3; 5;19; 43; 58) should be updated with more relevant and recent literature.
  • Pag 3 Line 134: Some hydroxybenzoic acid and hydroxycinnamic acid contain more than seven and nine carbon atoms respectively.
  • Pag 23 Line 672: Curcumin is a derivative of ferulic acid.
  • Table 1: Pomegranate extracts and coffee are not compound/ derivatives but complex mixture of compounds.
  • Table 2: Specify for soy protein which Isoflavones are more active.
  • Table 3: Add columns and rows
  • Table 4: Schisandrae Fructus extracts are not compound/ derivatives
  • Editing of bibliographic references is required (e.g. Ref. 3; 7; 9; 11…)

Reviewer 2 Report

The authors conducted a well-documented review, easy to understand for a larger group of professionals, not necessarily in the same field. The information presented in the review provides explanations regarding the beneficial mechanisms of some polyphenols as a prevention strategy for muscle disorders.

1) The only aspect I suggest to be improved is the conclusion of the review.

Could you consider customizing the conclusion by focusing on the polyphenols mentioned above and how they might benefit the treatment of muscle atrophy?

 2) It would also be interesting to mention in the discussion if clinical trials in this direction have been attempted and include a subheading mentioning natural polyphenols used in clinical trials to treat muscle disorders.

3) Give format or insert the correct table in table number 3.

4) In paragraph 465, consider rephrasing the sentence "reducing oxidative species" to “reducing ROS production”

Reviewer 3 Report

The review done by T. Nikawa and cols. Entitled “Polyphenols ant their effects on muscle atrophy and muscle health”, is a very interesting work in which the authors made a depth insight into the protective effects of several polyphenols on skeletal muscle in the setting of disease/damage, and conditions as aging. The basic methodology of the review, was in essence a deep search on database such as PubMed and Google scholar using keywords related to polyphenols and skeletal muscle, which was well implemented and developed. The search of topics related to skeletal muscle atrophy and polyphenols was well conducted and detailed described. However, there are a few topics (i.e. inflammation, oxidative stress) that were lightly addressed, but since their importance in the development of muscle atrophy it would be important to describe them in more detail along with the positive effects of PPs.

Comments:

Is there any evidence on skeletal muscle atrophy chemical compounds-induced? Further than steroids or statins, i.e. pesticides? please comment on this.

Please comment on the toxic effects evidence or experimental data showing no toxic effects of PPs

It would be helpful and quite descriptive if authors include a diagram showing the PPs family members more representative with impact on skeletal muscle.

The relevant work done by Ramirez-Sanchez et al Int J Cardiol 2013 and Taub P et al Clin Sci. 2013 describe the mechanistic pathway in which EC could be regulating antioxidant machinery in skeletal muscle of diabetic patients as well as the recovery of sarcomere structure. It would be interesting if authors comment on these topics.

Authors need to make a statement/paragraph on the current therapeutic strategies implemented targeting skeletal muscle atrophy
